# Long-Term Survival and Recurrence in Oropharyngeal Squamous Cell Carcinoma in Relation to Subsites, HPV, and p16-Status

**DOI:** 10.3390/cancers13112553

**Published:** 2021-05-23

**Authors:** Malin Wendt, Lalle Hammarstedt-Nordenvall, Mark Zupancic, Signe Friesland, David Landin, Eva Munck-Wikland, Tina Dalianis, Anders Näsman, Linda Marklund

**Affiliations:** 1Department of Clinical Science, Intervention and Technology—CLINTEC Division of Ear, Nose and Throat Diseases, Karolinska Institutet, Karolinska University Hospital, 171 64 Stockholm, Sweden; Lalle.Hammarstedt-Nordenvall@sll.se (L.H.-N.); David.Landin@ki.se (D.L.); Eva.Munck-Wikland@ki.se (E.M.-W.); Linda.Marklund@ki.se (L.M.); 2Medical Unit Head Neck, Lung and Skin Cancer, Karolinska University Hospital, 171 76 Stockholm, Sweden; Mark.Zupancic@ki.se (M.Z.); signe.friesland@sll.se (S.F.); Tina.Dalianis@ki.se (T.D.); 3Department of Oncology-Pathology, Karolinska Institutet, Bioclinicum J6:20, Karolinska University Hospital, 171 64 Stockholm, Sweden; Anders.Nasman@ki.se; 4Department of Clinical Pathology, CCK R8:02, Karolinska University Hospital, 171 64 Stockholm, Sweden

**Keywords:** human papillomavirus, HPV, oropharyngeal cancer, OPSCC, tonsillar cancer, survival, OS, DFS, p16, recurrence

## Abstract

**Simple Summary:**

Long-term survival in patients with oropharyngeal cancer is sparsely studied, but atypical recurrences in human papillomavirus-positive (HPV+) oropharyngeal cancer have been indicated. Furthermore, while the role of HPV is well established in tonsillar and base of tongue cancer, the dominant oropharyngeal subsites, its role in the minor oropharyngeal sites (the oropharyngeal walls, the uvula, and the soft palate) is not fully elucidated. The aim of this retrospective study was therefore to assess long-term outcome in relation to oropharyngeal sub-sites and HPV/p16 status. We confirm the prognostic role of p16+ in tonsillar and base of tongue cancer, but not the other sites. We find that combined HPV/p16-status gives better prognostic information than p16 alone. Lastly, we show that p16− cancer has more locoregional and late recurrences compared to p16+ cancer. Consequently, only combined HPV/p16 positivity in patients with tonsillar and tongue base cancer should be used in future treatment de-escalation trials.

**Abstract:**

Long-term survival data in relation to sub-sites, human papillomavirus (HPV), and p16^INK4a^ (p16) for patients with oropharyngeal squamous cell carcinoma (OPSCC) is still sparse. Furthermore, reports have indicated atypical and late recurrences for patients with HPV and p16 positive OPSCC. Therefore, we assessed long-term survival and recurrence in relation to oropharyngeal subsite and HPV/p16 status. A total of 529 patients with OPSCC, diagnosed in the period 2000–2010, with known HPVDNA and p16-status, were included. HPV/p16 status and sub-sites were correlated to disease-free and overall survival (DFS and OS respectively). The overexpression of p16 (p16^+^) is associated with significantly better long-term OS and DFS in tonsillar and base of tongue carcinomas (TSCC/BOTSCC), but not in patients with other OPSCC. Patients with HPVDNA^+^/p16^+^ TSCC/BOTSCC presented better OS and DFS compared to those with HPVDNA^−^/p16^−^ tumors, while those with HPVDNA^−^/p16^+^ cancer had an intermediate survival. Late recurrences were rare, and significantly more frequent in patients with p16^−^ tumors, while the prognosis after relapse was poor independent of HPVDNA^+/−^/p16^+/−^ status. In conclusion, patients with p16^+^ OPSCC do not have more late recurrences than p16^−^, and a clear prognostic value of p16^+^ was only observed in TSCC/BOTSCC. Finally, the combination of HPVDNA and p16 provided superior prognostic information compared to p16 alone in TSCC/BOTSCC.

## 1. Introduction

Already in 1983, Syrjänen et al. suggested a possible correlation between human papillomavirus (HPV) infection and head and neck squamous cell carcinoma (HNSCC) [1]. In the following decades, extensive research, by ourselves and others, established high-risk HPV infection as a risk factor of oropharyngeal squamous cell carcinoma (OPSCC), and especially tonsillar and base of tongue cancer (TSCC and BOTSCC) [2,3,4,5,6,7]. Data also showed that patients with HPV positive OPSCC, more specifically HPV positive TSCC and BOTSCC, had a better clinical outcome, as well as a different epidemiological profile, when compared to patients with corresponding HPV negative cancer [8,9,10]. More specifically, the incidence of HPV positive TSCC and BOTSCC, and thereby OPSCC, has continuously increased in the Western world since the 1970’s, and this increase has been described as an epidemic of viral induced OPSCC, or more specifically, TSCC and BOTSCC [5,11,12]. At the same time, the prevalence of smoking has decreased, resulting in a parallel decrease in the total incidence of HNSCC, but with a shift towards a larger proportion of HPV positive tumors in the oropharyngeal subsites [11].

The overexpression of p16^Ink4a^ (p16^+^) has, similar to the presence of HPV-DNA (HPVDNA^+^), been shown to have a strong correlation to active HPV infection (i.e., expression of HPV E6 mRNA) in OPSCC, although the combined presence of both p16^+^ and HPVDNA^+^ is superior to using these markers separately [13]. Because p16^+^ is easier to determine by immunohistochemistry (IHC), it was therefore suggested as a possible surrogate marker for HPV infection, and thereby also shown to be associated with a better prognosis in OPSCC [14,15]. In addition, p16^+^ as a marker of HPV infection is now used in the 8th version of the American Joint Committee of Cancer (AJCC) AJCC Staging manual (TNM-8) to separate HPV-related from HPV-unrelated OPSCC [16].

However, recent studies, by ourselves and others, have suggested that the prevalence of HPV infection, its correlation to p16^+^, and the impact of HPV infection on prognosis differs considerably between OPSCC sites. More specifically, the role of HPV on survival differs between tumors arising in lymphoepithelial oropharyngeal sites, i.e., TSCC and BOTSCC, and carcinomas arising in non-lymphoepithelial subsites of the oropharynx, i.e., carcinomas of the soft palate, uvula, and posterior pharyngeal wall (otherOPSCC) [17]. Therefore, while the prognostic role of HPV and the correlation between HPV infection and p16^+^ is established in TSCC and BOTSCC, the role of HPV and its correlation to p16^+^ in otherOPSCC is more ambiguous [18,19,20,21,22,23,24].

Irrespectively, roughly 10–20% of all patients with p16^+^ or HPVDNA^+^ OPSCC develop recurrent disease within 5 years after diagnosis [25,26]. Moreover, previous observations have proposed that patients with p16^+^ or HPVDNA^+^ OPSCC exhibit a different pattern of recurrence compared to those with corresponding p16^−^ or HPVDNA^−^ OPSCC [27,28]. It has specifically been indicated that the former group presents later recurrences and at different sites, compared to the latter, an issue that needs further study [29,30].

The aim of the present study was therefore to accumulate more knowledge regarding long-term survival, and recurrence in patients with OPSCC in relation to p16^+^ and HPVDNA^+^ status and OPSCC subsite.

## 2. Materials and Methods

### 2.1. Patients’ Characteristics

All patients diagnosed in the period 2000–2010 with OPSCC, (TSCC: ICD-10 C09.0-9 and C02.4; BOTSCC C01.9; otherOPSCC: C10.0-9, C05.1-9), in the County of Stockholm/Gotland, Sweden were identified through the Swedish Cancer Registry. Patients treated with palliative intent were excluded from further studies. In addition, patients diagnosed with only cytology and/or with unknown p16 status, or there were no biopsies available, were also excluded (n = 25), and of these, 4 patients had a recurrence: 3 locoregional relapse (LRR) and 1 distant relapse (DR). Consequently, the study base consisted of 529 OPSCC patients, treated with the intention to cure and with known p16 and HPV DNA status (see below).

Patients’ charts were assessed for TNM 7-stage, age, gender, smoking, WHO-status, type of treatment, recurrence, time to recurrence and location of recurrence, and survival. Treatment was classified as surgery, radiotherapy (including both external and brachytherapy), or chemoradiotherapy. Smoking data was obtained whenever noted in the charts, and was classified as “never” smoker or as “ever” smoker. The study was conducted according to ethical permissions 2005/431-31/4, 2005/1330-32 and 2009/1278-31/2 from the Stockholm Regional Ethical Review Board.

### 2.2. HPV-DNA and Overexpression of p16

Data on HPV DNA status and p16 status were obtained from previous studies [19,31]. Briefly, DNA was extracted from pre-treatment FFPE biopsies and analyzed for the presence of HPV DNA by PCR using broad-spectrum general primers bsGP5+/6+, and HPV-typing was performed utilizing a bead based multiplex assay (Luminex Magpix; Austin, TX, USA). For details, see [19,31].

Likewise, data on p16^Ink4a^ overexpression by immunohistochemistry was obtained from previous studies [18,32]. Overexpression of p16 (p16^+^) was defined as a strong nuclear and cytoplasmatic staining in more than 70% of tumor cells, as suggested by the College of American Pathologists [33].

### 2.3. Statistical and Survival Analysis

To evaluate differences in categorical data, we used a Chi-^2^ test; for the continuous variables we used an independent two-tailed *t*-test.

Overall survival (OS) was defined as time from diagnosis until death of any cause. Disease-free survival (DFS) was defined as time from diagnosis until recurrence of disease. A patient was considered to have recurrence of disease when treatment with curative intent was completed, patient was assessed as complete response at check-up, and then having a recurrence confirmed by radiology and/or histopathology. Recurrence was classified as LRR if in T- or N-position, and DR in M-position. Time-to recurrence was calculated as time from diagnosis until confirmed recurrence was noted in the patient chart. Patients who died tumor-free were censored at the time of death. Patients that never became tumor-free after treatment were censored at day 0. All patients had clinical controls every 3 months for the first 2 years, then every 6 months for a total of 5 years, and then if patients showed symptoms. In the case of recurrent disease, patients started clinical controls according to the same schedule after treatment.

OS and DFS was calculated for the whole OPSCC group, as well as for TSCC/BOTSCC and otherOPSCC separately, in relation to p16. Outcome was set as death of any cause or recurrence (LRR or DR). Results were presented in Kaplan–Meier curves, and survival was assessed with a log-rank test.

A sub-group analysis with univariate and multivariate analyzes was performed in patients with TSCC/BOTSCC and performance status (WHO/ECOG) (PS) 0 as a proxy for capacity to fulfill treatment. Here, hazard ratios (HR) for the combinations of HPVDNA^+/−^/p16^+/−^ status, age, stage, and smoking were estimated using the Cox proportional hazard model.

All analyses were made in SPSS (version 25 for Mac); *p*-values of <0.05 were considered significant.

## 3. Results

### 3.1. Patient and Tumor Characteristics

All patients at baseline, and their tumor characteristics, are depicted in Table 1. The largest proportion of primary tumors were TSCC (63%) and BOTSCC (22%), followed by otherOPSCC (15%). Patients with TSCC and BOTSCC (TSCC/BOTSCC) were younger at diagnosis compared to otherOPSCC (60.8 vs. 65.2 years, *p* = 0.01), and were diagnosed with a more advanced N-status (N0-1 vs. N2-3, *p* = 0.01).

TSCC/BOTSCC were significantly more frequently p16^+^ and HPV DNA^+^, compared to otherOPSCC (70.4% p16^+^ TSCC/BOTSCC vs. 22.2% p16^+^ otherOPSCC, *p* < 0.0001, and 72.5% HPVDNA^+^ TSCC/BOTSCC vs. 21% HPVDNA^+^ otherOPSCC, *p* < 0.0001).

In the whole OPSCC cohort, 110/529 patients (20.8%), had recurrent disease, more specifically with 17.8% recurrences in TSCC, 24.2% in BOTSCC, and 28.8% in otherOPSCC (Table 1).

### 3.2. Long-Term Overall Survival and Disease-Free Survival in Relation to p16 and Subsites

Patients with p16^+^ OPSCC had a significantly better OS and DFS, compared to patients with p16^−^ OPSCC (log rank test <0.0001 and <0.0001, respectively). More specifically, 5-year OS and DFS were 76.4% and 84.9%, respectively, in patients with p16^+^ OPSCC, and 42.4% and 64.7%, respectively, in patients with p16^−^ OPSCC (Figure 1A,B). The 10-year OS and DFS were 65.4% and 83.7%, respectively, in patients with p16^+^ OPSCC, and 22.9% and 59.6%, respectively, in patients with p16^−^ OPSCC (Figure 1A,B).

When analyzing survival per subsite, patients with p16^+^ TSCC/BOTSCC had a significantly better OS and DFS compared to patients with p16^−^ TSCC/BOTSCC (log rank test < 0.0001 and <0.0001, respectively). More specifically, 5-year OS and DFS were 77.4% and 84.7%, respectively, in patients with p16^+^ TSCC/BOTSCC, compared to 43.4% and 65.4%, respectively, in patients with p16^−^ TSCC/BOTSCC. The 10-year OS and DFS were 66.3% and 83.8%, respectively, in patients with p16^+^ TSCC/BOTSCC, compared to 27% and 59.7%, respectively, in patients with p16^−^ TSCC/BOTSCC (Figure 1C,D). Notably, few late recurrencies (>5 years after diagnosis) were observed in patients with p16^+^ TSCC/BOTSCC, which was not entirely the case in patients with p16^−^ TSCC/BOTSCC (Figure 1D). Patients with p16^−^ TSCC/BOTSCC had, in fact, a significantly higher risk of having a recurrent disease five years after diagnosis, compared to patients with p16^+^ TSCC/BOTSCC (*p* < 0.001, Figure 2).

No differences in OS or DFS with regard to p16^+/−^ status were observed in patients with otherOPSCC (Figure 1E,F).

### 3.3. Localisation of Recurrence and Relation to Overexpression of p16 and Subsites

Of 112 recurrences, 74 (66%) were LRR and 38 (34%) were DR. In TSCC/BOTSCC, there was an equal distribution of LRR and DR, while most otherOPSCC had LRR (Figure 3). Notably, patients with p16^+^ OPSCC had a significantly higher proportion of DR, compared to those with p16^−^ OPSCC (p16^+^ OPSCC: Local relapse: *n* = 16; Regional relapse: *n* = 15; DR: *n* = 24; p16^−^ OPSCC: Local relapse: *n* = 37; Regional relapse: *n* = 6; DR: *n* = 14) (*p* = 0.03). Likewise, patients with p16^+^ TSCC/BOTSCC tended to have more DR compared to those with p16^−^ TSCC/BOTSCC, while the opposite was observed in patients with otherOPSCC; however, neither of these latter trends were significant.

### 3.4. Long-Term Overall and Disease-Free Survival in Relation to Both Overexpression of p16 and Presence of HPV DNA, in Patients with TSCC/BOTSCC

Because the prognostic role of p16^+^ alone was only observed in patients with TSCC/BOTSCC, we continued our analysis by adding the presence of HPV DNA (HPVDNA^+^) as an additional prognostic marker, only in these tumors.

Patients with combined HPVDNA^+^/p16^+^ TSCC/BOTSCC had a significantly better OS and DFS compared to patients with HPVDNA^−^/p16^−^ TSCC/BOTSCC (log rank: *p* < 0.0001 and *p* < 0.0001). However, patients with HPVDNA^−^/p16^+^ TSCC/BOTSCC presented an intermediate survival (Figure 4A,B). More explicitly, patients with HPVDNA^−^/p16^+^ TSCC/BOTSCC had a better OS and DFS compared to patients with HPVDNA^−^/p16^−^ cancer (log rank test: *p* = 0.001 and *p* = 0.05, respectively), but a worse OS compared to patients with HPVDNA^+^/p16^+^ cancer (OS: log rank test: *p* = 0.047), and the trend was similar for DFS (log rank test: *p* = 0.1). Similarly, patients with HPVDNA^+^/ p16^−^ TSCC/BOTSCC (*n* = 42) also presented an intermediate survival.

In more detail, 5-year OS and DFS were 78.8% and 85.9%, respectively, in patients with HPVDNA^+^/p16^+^ TSCC/BOTSCC and 37.9% and 57.5%, respectively, in patients with HPVDNA^−^/p16^−^ cancer. Patients with discordant HPV and p16 status presented an intermediate survival, with 63.3% 5-year OS and 74.1% 5-year DFS in those with HPVDNA^−^/p16^+^ TSCC/BOTSCC, while corresponding figures for those with HPVDNA^+^/p16^−^ cancers were 57.1% and 85.4% for OS and DFS, respectively. Moreover, 10-year OS and DFS were 67.8% and 85.6%, respectively, in patients with HPVDNA^+^/p16^+^ TSCC/BOTSCC and 11.5% and 50.9%, respectively, in patients with HPVDNA^−^/p16^−^ cancer. Patients with discordant HPV and p16 status presented an intermediate survival, with 36.5% 10-year OS and 74.1% DFS in those with HPVDNA^−^/p16^+^ TSCC/BOTSCC, while corresponding figures for those with HPVDNA^+^/p16^−^ cancers were 42.9% and 80.9% for OS and DFS, respectively.

### 3.5. HPV, p16 Status and Other Prognostic Factors in Patients with TSCC/BOTSCC

A univariate and multivariate subgroup analysis, including patients with TSCC/BOTSCC and performance status (WHO/ECOG) (PS) 0, as a surrogate marker for completion of intended treatment, was performed and included 87.7% of all TSCC/BOTSCC patients. Uni- and multi-variate analyses for OS and DFS were performed for HPVDNA^+/−^/p16^+/−^ status, age, dichotomized smoking status (Ever vs. Never), and dichotomized TNM-7 stage (1–2 vs. 3–4) (Table 2).

Notably, patients with HPVDNA^+/^p16^+^ TSCC/BOTSCC had a clearly better OS and DFS than those with HPVDNA^−^/p16^−^ TSCC/BOTSCC, both in the uni- and in the multivariate analysis, while those with HPVDNA^−^/p16^+^ TSCC/BOTSCC only had a significantly better OS compared to those with HPVDNA^−^/p16^−^ cancer in the univariate analysis (Table 2). Moreover, irrespective of HPVDNA and p16, age was also significantly correlated to both OS and DFS in the univariate and multivariate analysis. For the dichotomized stage (1–2 vs. 3–4), no significant differences were observed for either OS or DFS (Table 2).

### 3.6. Survival after Recurrence TSCC/BOTSCC in Relation to HPV and Overexpression of p16

Survival after recurrence (LRR or DR) in patients with TSCC/BOTSCC was also assessed in correlation to HPVDNA^+/−^ and p16^+/−^ status. Survival was generally low (5.9%) after LRR/DR and did not differ significantly between patients with HPVDNA^+^/p16^+^, HPVDNA^−^/p16^−^, and HPVDNA^−^/p16^+^ TSCC/BOTSCC (Figure 5).

## 4. Discussion

In this large, long term follow-up cohort study of OPSCC and its subsites, we disclosed that p16^+^ was correlated to a favorable OS and DFS in patients with TSCC/BOTSCC as compared to those with corresponding p16^−^ cancer, while in patients with otherOPSCC, no such analogy was observed. Likewise, patients with HPVDNA^+^/p16^+^ TSCC/BOTSCC presented a better OS and DFS compared to those with HPVDNA^−^/p16^−^ tumors, while notably, those with HPVDNA^−^/p16^+^ carcinomas presented an intermediate survival. Finally, late recurrences were rare, but were significantly more frequent in patients with p16^−^ tumors, but nevertheless, the prognosis for recurrent disease was poor independent of the HPVDNA^+/−^/p16^+/−^ status of the tumor.

Consequently, in this report, we confirm previous results by ourselves and others that patient and tumor characteristics differ significantly between OPSCC arising in a lymphoepithelial context (TSCC/BOTSCC) and those arising in a non-lymphoepithelial context (otherOPSCC) [17,18,19,20,21,22,23,24]. In a previous study, we found that p16-status was poorly correlated to 5-year OS and DFS in patients with otherOPSCC, and that p16^+^ was not a reliable surrogate marker for active HPV infection in this tumor type [32]. In addition, we have earlier shown that patients with p16^+^ otherOPSCC at a low stage have a significantly worse OS compared to patients with TSCC/BOTSCC at the same stage [18]. Taken together, the data suggest that survival and pattern of recurrence for p16^+^ otherOPSCC, instead resembles p16^−^ TSCC/BOTSCC/otherOPSCC. Consequently, p16^+^ status is not suitable to determine stage or choices of treatment in patients with otherOPSCC.

In addition, we also demonstrated in a subgroup analysis of TSCC/BOTSCC patients that patients with discordant (HPVDNA^−^/p16^+^) have a worse survival, compared to those with HPVDNA^+^/p16^+^ TSCC/BOTSCC. Moreover, patients with discordant HPV and p16 status had a more ambiguous survival benefit over patients with HPVDNA^−^/p16^−^ cancer, compared to those with HPVDNA^+^/p16^+^ TSCC/BOTSCC. Our data imply that there may be a risk of future undertreatment of patients with HPVDNA^−^/p16^+^ tumors, and possibly the opposite upon treatment of patients with HPVDNA^+^/p16^−^ tumors when patients are classified only based on p16^+/−^ status with the new TNM- 8 staging system. These data suggest that the prognostic value of p16 is inferior to the combination of p16 and HPVDNA [13].

Although the above data need to be confirmed, they still suggest that some caution is necessary when conducting novel treatment strategies based on the new TNM-8 staging system [20].

Finally, we could not confirm earlier findings from smaller studies that patients with p16^+^ TSCC/BOTSCC have a higher incidence of late relapses compared to patients with p16^−^ TSCC/BOTSCC [32,33]. In general, only few cases relapsed after 5 years, and these tended to originate from p16^–^ tumors. In addition, we did not observe any difference in survival between patients with p16^+^ and p16^−^ TSCC/BOTSCC-patients upon recurrence, which also differs from some earlier reports [34,35].

Of note, in most countries, the current first line of treatment in OPSCC is radiotherapy or chemoradiotherapy, with fairly good results in patients with p16^+^ TSCC/BOTSCC, but with less favorable outcome in patients with p16^−^ TSCC/BOTSCC and with otherOPSCC, regardless of p16 status. This calls for further studies on how to improve survival for patients with p16^−^ TSCC/BOTSCC. As for otherOPSCC, it is not unlikely that the tumors within this group may better resemble oral cancer and benefit from primary surgery regardless of p16^+/−^ status. In fact, one report demonstrated better recurrence free survival in patients with p16^−^ OPSCC when they were treated with upfront surgery instead of radiotherapy and chemotherapy; however, surgery did not have an impact on OS [36].

This study has several limitations. Firstly, although the study population is relatively large, it is a retrospective single-center study with clinical data collected prospectively from patients’ records. Secondly, we have not adjusted data for treatment modalities or for changes in treatment regimens over time. However, at our center, treatment has been consistent within and between the subsites over time, irrespective of p16^+/−^ and HPVDNA^+/−^ status, implying accuracy in the results. Finally, the group of otherOPSCC was relatively small, resulting in limited numbers of p16^+^ tumors in that group. To confirm our findings, we encourage larger multicenter studies.

## 5. Conclusions

In conclusion, this study of long-term outcome and recurrence shows that patients with p16^+^ TSCC/BOTSCC have better long-term OS and DFS than patients with corresponding p16^−^ cancer. However, the combination of HPVDNA^+^ and p16^+^ status presented more accurate and detailed prognostic information than p16^+/−^ status alone in patients with TSCC/BOTSCC. We therefore recommend the use of combined HPVDNA^+^/p16^+^ analysis for TSCC/BOTSCC when conducting treatment decisions and future tailored trials. In addition, importantly, p16^+^ status did not affect long-term outcome in patients with otherOPSCC. However, larger studies will be required to confirm these results. Finally, late recurrences were rare in OPSCC, and patients with p16^+^ TSCC/BOTSCC did not have a higher frequency of late metastasis compared to those with corresponding p16^−^ carcinomas, nor did p16^+^ status in their carcinomas affect outcome after recurrence.

## Figures and Tables

**Figure 1 cancers-13-02553-f001:**
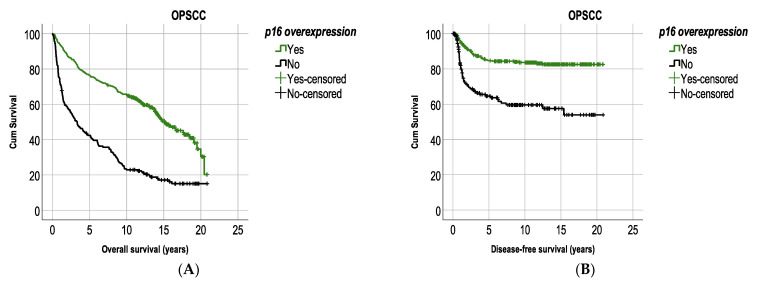
Kaplan–Meier figures with overall survival (OS) and disease-free survival (DFS) in patients with OPSCC (**A**,**B**) and separated on subsite with TSCC/BOTSCC (**C**,**D**) and otherOPSCC (**E**,**F**). (**A**,**B**) Patients with p16 positive OPSCC had a significantly better OS and DFS compared to patients with p16 negative OPSCC (log rank: *p* < 0.0001 and *p* < 0.0001, respectively). (**C**,**F**) Patients were separated into those with cancer in lymphoepithelial sub-sites (TSCC/BOTSCC) and non-lymphoepithelial subsites (otherOPSCC) and analyzed separately. (**C**,**D**) Patients with p16 positive TSCC/BOTSCC had a significantly better OS and DFS compared to patients with p16 negative TSCC/BOTSCC (log rank: *p* < 0.0001 and *p* < 0.0001, respectively). (**E**,**F**) No significant differences in OS and DFS between patients with p16 positive and p16 negative otherOPSCC were observed (log rank: *p* = 0.13 and *p* = 0.9, respectively).

**Figure 2 cancers-13-02553-f002:**
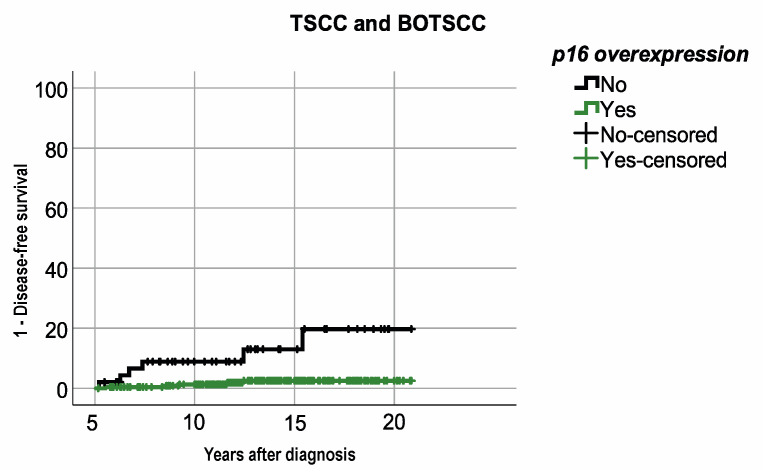
Kaplan–Meier figure with 1-disease-free survival (DFS) in patients with TSCC/BOTSCC five years after diagnosis. Patients with p16 positive TSCC/BOTSCC had a significantly better DFS compared to patients with p16 negative TSCC/BOTSCC (log rank: *p* < 0.001).

**Figure 3 cancers-13-02553-f003:**
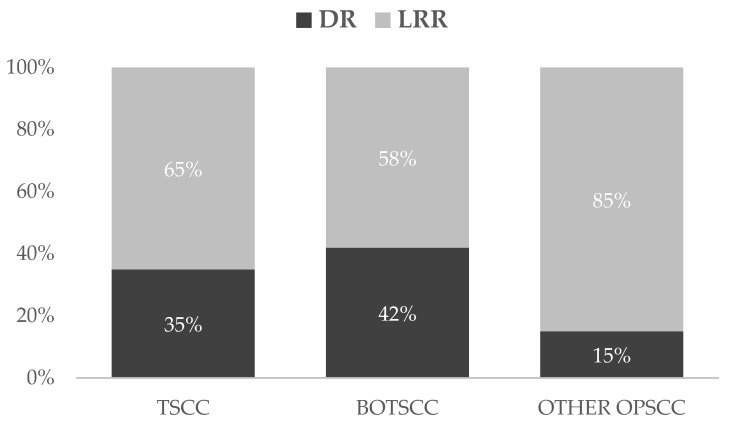
Distribution (percent) of loco-regional (LRR) and distant (DR) recurrencies separated per OPSCC sub-site. Patients with TSCC and BOTSCC had more often DR compared to otherOPSCC (*TSCC*: DR: *n* = 21, LRR: *n* = 39; *BOTSCC*: DR *n* = 13, LRR *n* = 18; otherOPSCC: DR *n* = 3, LRR *n* = 17).

**Figure 4 cancers-13-02553-f004:**
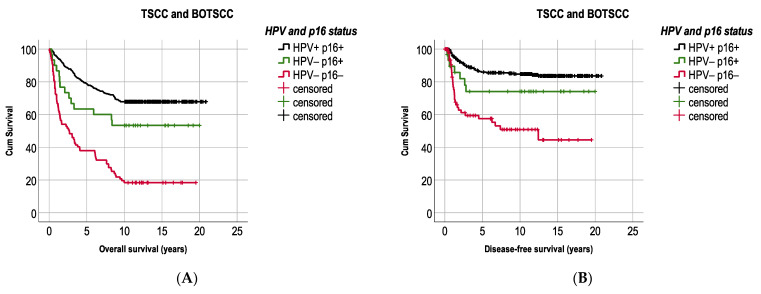
Kaplan–Meier figures with overall survival (OS) and disease-free survival (DFS) in patients with TSCC/BOTSCC (**A**,**B**). Patients with HPV DNA positive and p16 positive (HPV+/p16+) TSCC/BOTSCC had a significantly better OS and DFS, respectively (**A**,**B,** respectively), as compared to patients with HPV DNA negative and p16 negative TSCC/BOTSCC (log rank: *p* < 0.0001 and *p* < 0.0001, respectively). However, patients with p16 positive but HPV negative (HPV−/p16+) TSCC/BOTSCC presented an intermediate OS and DFS compared to patients with double positive or double negative HPV/p16 status. (HPV+p16+ vs. HPV−p16+ (log rank test): OS: *p* = 0.047; DFS: *p* = 0.1, and HPV−p16+ vs. HPV−p16− (log rank test): OS: *p* = 0.001; DFS: *p* = 0.05).

**Figure 5 cancers-13-02553-f005:**
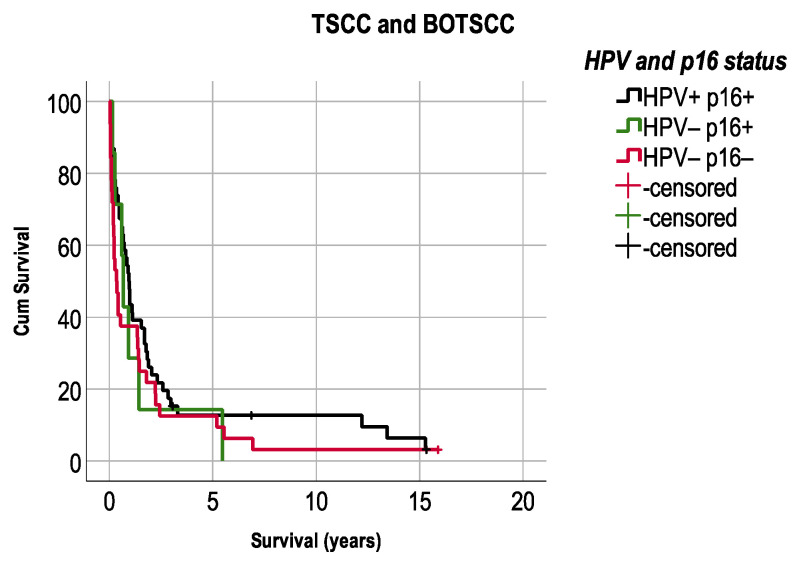
Kaplan–Meier figures with survival after a loco-regional or distant recurrence (LRR/DR) separated for HPV and p16 status. No significant differences were observed in survival between patients with TSCC/BOTSCC after recurrent disease independent of HPV and p16 status. (HPV+p16+ vs. HPV−p16− (log rank test): *p* = 0.17).

**Table 1 cancers-13-02553-t001:** Patient and tumor characteristics.

		TSCC (%)	BOTSCC (%)	OtherOPSCC (%)	Total (%)
Number of patients		337	126	66	529
Age	Mean	60.4	62	65.2	61
	Median	59	62	65	61
	Range	29–90	30–84	46–88	29–90
Sex	Female	82 (24%)	39 (31%)	23 (35%)	144 (27%)
	Male	255 (76%)	87 (69%)	43 (65%)	385 (73%)
T	T1	78 (23%)	40 (32%)	8 (12%)	126 (24%)
(AJCC 7th Edition)	T2	129 (38%)	33 (26%)	24 (36%)	186 (35%)
	T3	70 (21%)	17 (13%)	25 (38%)	112 (21%)
	T4a	56 (17%)	36 (29%)	7 (11%)	99 (19%)
	T4b	4 (1%)	0 (0%)	2 (3%)	6 (1%)
N	N0	70 (21%)	28 (22%)	29 (44%)	127 (24%)
(AJCC 7th Edition)	N1	72 (21%)	16 (13%)	12 (18%)	100 (19%)
	N2a	49 (15%)	13 (10%)	4 (6%)	66 (12%)
	N2b	112 (33%)	32 (25%)	9 (14%)	153 (29%)
	N2c	21 (6%)	27 (21%)	11 (17%)	59 (11%)
	N3	13 (4%)	7 (6%)	1 (2%)	21 (4%)
	NX	0 (0%)	3 (2%)	1 (2%)	3 (1%)
M	M0	335 (99%)	120 (95%)	63 (95%)	518 (98%)
(AJCC 7th Edition)	M1	2 (1%)	2 (2%)	0 (0%)	4 (1%)
	MX	0 (0%)	4 (3%)	3 (5%)	7 (1%)
TNM Stage	I	10 (3%)	8 (6%)	3 (21%)	21 (4%)
(AJCC 7th Edition)	II	25 (7%)	7 (6%)	12 (21%)	44 (8%)
	III	83 (25%)	20 (16%)	23 (15%)	126 (24%)
	IVa	198 (59%)	82 (65%)	25 (33%)	305 (58%)
	IVb	19 (6%)	7 (6%)	3 (6%)	29 (5%)
	IVc	2 (1%)	2 (2%)	0 (4%)	4 (1%)
Smoking	Ever	237 (70%)	79 (63%)	51 (77%)	367 (69%)
	Never	92 (27%)	43 (34%)	7 (11%)	142 (27%)
	Not known	8 (2%)	4 (3%)	4 (6%)	20 (4%)
p16 overexpression	No	81 (24%)	41 (33%)	60 (91%)	182 (34%)
	Yes	256 (76%)	85 (67%)	6 (9%)	347 (66%)
HPV DNA status	Negative	78 (23%)	39 (31%)	47 (71%)	164 (31%)
	Positive	259 (77%)	87 (69%)	15 (23%)	361 (68%)
	Not known	0 (0%)	0 (0%)	4 (6%)	4 (1%)
Performance status	0	304 (90%)	102 (81%)	27 (41%)	433 (82%)
(WHO/ECOG)	1	20 (6%)	19 (15%)	20 (30%)	59 (11%)
	2	7 (2%)	5 (4%)	16 (24%)	28 (5%)
	3	5 (1%)	0 (0%)	3 (5%)	8 (2%)
	Not known	1 (0%)	0 (0%)	0 (0%)	1 (0%)
Treatment ^1^	RT	234 (69%)	56 (44%)	48 (73%)	340 (64%)
	CRT	101 (30%)	70 (56%)	17 (26%)	183 (35%)
	Primary surgery	2 (1%)	0 (0%)	4 (6%)	6 (1%)
Recurrence	Yes	61 (18%)	30 (24%)	19 (29%)	110 (21%)
	No	276 (82%)	96 (76%)	47 (71%)	419 (79%)

^1^ Neck dissection not included.

**Table 2 cancers-13-02553-t002:** Uni- and multivariable analysis of OS and DFS in patients with TSCC/BOTSCC and PS 0.

		Overall Survival (OS)						Disease-Free Survival (DFS)					
		Univariable			Multivariable			Univariable			Multivariable		
		HR	95% CI	*p*-Value	HR	95% CI	*p*-Value	HR	95% CI	*p*	HR	95% CI	*p*
HPV/p16 status	HPV− p16−	1			1			1			1		
	HPV− p16+	0.31	0.15–0.66	0.002	0.49	0.23–1.05	0.07	0.34	0.12–0.98	0.05	0.43	0.15–1.3	0.1
	HPV+ p16+	0.23	0.17–0.34	<0.0001	0.30	0.21–0.44	<0.0001	0.25	0.15–0.42	<0.0001	0.29	0.17–0.50	<0.0001
Age		1.06	1.05–1.09	<0.0001	1.1	1.04–1.08	<0.0001	1.04	1.02–1.06	0.001	1.03	1.008–1.06	0.009
Smoking	Ever	1			1			1			1		
	Never	0.46	0.30–0.68	<0.0001	0.58	0.38–0.88	0.01	0.54	0.31–0.93	0.03	0.69	0.39–1.3	0.2
Stage (TNM-7)	I/II	1			1			1			1		
	III/IV	1.12	0.66–1.9	0.7	1.6	0.93–2.8	0.09	1.36	0.59–3.1	0.5	1.8	0.78–4.3	0.2

## Data Availability

The data presented in this study are available on request from the corresponding author. The data are not publicly available due to Swedish laws on personal confidential information.

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
