# Peer review of "Long-Term Survival and Recurrence in Oropharyngeal Squamous Cell Carcinoma in Relation to Subsites, HPV, and p16-Status"

_cancers, 2021, doi:10.3390/cancers13112553_

Round 1
Reviewer 1 Report
It is well written and interesting manuscript. My only coment is to claryfi the statement from 27 line about local recurrences. I did not find any data in the text considering the ratio of local failures. All failures were given together with regional onces. Otherwise, it would be interesting to know the ratio of local and nodal failures separaterly taking into account more advanced nodal disease in HPV+ patients. Such observation also may indciate what kind of follow-up examination (more local or more nodal) should be carried on.
Author Response
We thank the reviewers for their valuable comments and have revised the manuscript accordingly. In addition, the language has also been revised.
Reviewer 1
It is well written and interesting manuscript.
My only coment is to claryfi the statement from 27 line about local recurrences. I did not find any data in the text considering the ratio of local failures. All failures were given together with regional onces. Otherwise, it would be interesting to know the ratio of local and nodal failures separaterly taking into account more advanced nodal disease in HPV+ patients. Such observation also may indciate what kind of follow-up examination (more local or more nodal) should be carried on.
We thank the reviewer for carefully reviewing the text and for valuable input. We have now corrected the typo on line 27 to “locoregional”. In addition, we have now also specified local and regional failures separately in section 3.3, lines 195-199.
Reviewer 2 Report
This is a straightforward set of analyses addressing the utility of HPV biomarkers in prediction of recurrence and survival in a cohort from Sweden, being particularly attentive to orophyaryngeal subsite (tonsil/base of tongue). While these analyses are not terribly novel, the sample size of 529 tumors is substantial in comparison to other studies, with a depth of information (p16 staining, HPV status, HPV DNA) that facilitates some subgroup analyses that are a very useful contribution to the literature, as well as almost complete data for the variables. These subgroup analyses are quite useful as treatment options are considered for OPSCC. I have a few small comments that are easily addressable, but overall, these are clean and well-justified analyses and the manuscript is well-justified and clear.
- Would be useful to know the time frame from which the cancers were incident (years of diagnosis)
- Add age range to the demographic table
- I'd suggest that Figure 3 be denoted as % in the bars, with total N in the legend (small point)
Author Response
We thank the reviewers for their valuable comments and have revised the manuscript accordingly. In addition, the language has also been revised.
Reviewer 2
This is a straightforward set of analyses addressing the utility of HPV biomarkers in prediction of recurrence and survival in a cohort from Sweden, being particularly attentive to oropharyngeal subsite (tonsil/base of tongue). While these analyses are not terribly novel, the sample size of 529 tumors is substantial in comparison to other studies, with a depth of information (p16 staining, HPV status, HPV DNA) that facilitates some subgroup analyses that are a very useful contribution to the literature, as well as almost complete data for the variables. These subgroup analyses are quite useful as treatment options are considered for OPSCC. I have a few small comments that are easily addressable, but overall, these are clean and well-justified analyses and the manuscript is well-justified and clear.
- Would be useful to know the time frame from which the cancers were incident (years of diagnosis)
We thank the reviewer for the comment. The time frame is given on line 93, i.e. from 2000-2010. Patients are usually followed for five years in the clinic, and then on demand if symptoms or other findings indicating relapse occurs. In addition, we have followed patients for at least ten years regarding date and cause of death, through the Swedish death registry.
- Add age range to the demographic table
We agree with the reviewer that demographical data is of importance and have now added age range and median age to Table 1 as suggested.
- I'd suggest that Figure 3 be denoted as % in the bars, with total N in the legend (small point)
We thank the reviewer for this comment and have now changed Figure 3 and the figure legend accordingly
Reviewer 3 Report
The article presents the results of a large-scale study, the undoubted merit of which is precisely in a representative sample. However, it bothers me that in addition to tonsillar and base of tongue carcinomas (TSCC / BOTSCC), the study included a small group of patients designated as other OPSCCs. The authors draw conclusions from this small generalized group, but within this group the results can be quite different. Therefore, in my opinion, it is necessary to add detailed information about which types of cancer are included and provide data on HPVDNA +/- and p16 +/- status. Only then can we conclude that it is legitimate to combine them into one subgroup. The data in the table show that there are only 6 patients with p16 + in the other OPSCC group, this must be explained in terms of the types of cancer included. There is no data on the relationship of overall and disease-free survival with the type of treatment in combination with HPVDNA +/- and p16 +/- status.
Author Response
We thank the reviewers for their valuable comments and have revised the manuscript accordingly. In addition, the language has also been revised.
Reviewer 3
The article presents the results of a large-scale study, the undoubted merit of which is precisely in a representative sample.
However, it bothers me that in addition to tonsillar and base of tongue carcinomas (TSCC / BOTSCC), the study included a small group of patients designated as other OPSCCs.
The authors draw conclusions from this small generalized group, but within this group the results can be quite different. Therefore, in my opinion, it is necessary to add detailed information about which types of cancer are included and provide data on HPVDNA +/- and p16 +/- status. Only then can we conclude that it is legitimate to combine them into one subgroup. The data in the table show that there are only 6 patients with p16 + in the other OPSCC group, this must be explained in terms of the types of cancer included.
We thank the reviewer for this comment. In line 94, we define the group entitled as “otherOPSCC” to be only squamous cell carcinomas arising in the non-lymphoepithelial parts of the oropharynx, i.e., the soft palate and uvula (C05.1-9) and the vallecula and pharyngeal walls (C10.0-9), in accordance with the definition in TNM 7 and TNM 8. Of the six patients with p16 positive tumors, five were also HPV DNA positive, and their localizations were in the pharyngeal walls (n = 4) and soft palate (n = 1).
We also agree with the reviewer that the “otherOPSCC” constitutes a small tumor group within the large group of oropharyngeal carcinomas. However, the subgroup of non-lymphoepithelial “otherOPSCC” show an internal homogeneity with important and unique characteristics in relation to the dominating lymphoepithelial subsites (TSCC and BOTSCC).[1-4] Therefore, we find it important to analyze them separately.
There is no data on the relationship of overall and disease-free survival with the type of treatment in combination with HPVDNA +/- and p16 +/- status.
We agree with the reviewer that the lack of detailed data on treatment is a weakness in this type of retrospective study. However, as depicted in the discussion line 316-319, treatment has been consistent within and between the subsites over time, irrespective of p16+/- and HPVDNA+/- status.
References
- Marklund, L.; Nasman, A.; Ramqvist, T.; Dalianis, T.; Munck-Wikland, E.; Hammarstedt, L. Prevalence of human papillomavirus and survival in oropharyngeal cancer other than tonsil or base of tongue cancer. Cancer Med 2012, 1, 82-88, doi:10.1002/cam4.2.
- Haeggblom, L.; Ramqvist, T.; Tommasino, M.; Dalianis, T.; Nasman, A. Time to change perspectives on HPV in oropharyngeal cancer. A systematic review of HPV prevalence per oropharyngeal sub-site the last 3 years. Papillomavirus Res 2017, 4, 1-11, doi:10.1016/j.pvr.2017.05.002.
- Tham, T.; Wotman, M.; Roche, A.; Kraus, D.; Costantino, P. The prognostic effect of anatomic subsite in HPV-positive oropharyngeal squamous cell carcinoma. Am J Otolaryngol 2019, 40, 567-572, doi:10.1016/j.amjoto.2019.05.006.
- Gelwan, E.; Malm, I.J.; Khararjian, A.; Fakhry, C.; Bishop, J.A.; Westra, W.H. Nonuniform Distribution of High-risk Human Papillomavirus in Squamous Cell Carcinomas of the Oropharynx: Rethinking the Anatomic Boundaries of Oral and Oropharyngeal Carcinoma From an Oncologic HPV Perspective. Am J Surg Pathol 2017, 41, 1722-1728, doi:10.1097/PAS.0000000000000929.
Round 2
Reviewer 3 Report
I have no more comments on the article. Considering that in its present form it can be recommended for publication.